# First Total Diet Study of Aflatoxins in Singapore: Exposure Risk, High-Risk Foods, and Public Health Implications

**DOI:** 10.3390/toxins17070324

**Published:** 2025-06-25

**Authors:** Ker Lew, Yu Lee Leyau, Ping Shen, Xin Li, Sherine Liew, Joachim Chua, Hui Yi Lim, Yuansheng Wu, Kern Rei Chng, Sheot Harn Chan

**Affiliations:** 1National Centre for Food Science, Singapore Food Agency, 7 International Business Park, Singapore 609919, Singapore; lew_ker@sfa.gov.sg (K.L.); leyau_yu_lee@sfa.gov.sg (Y.L.L.);; 2Department of Food Science & Technology, National University of Singapore, 2 Science Drive, Singapore 117543, Singapore

**Keywords:** aflatoxins, AFs, dietary exposure, estimated daily intake, EDI, margin of exposure, MOE, liver cancer risk, hazard quotient, HQ, total diet study, TDS

## Abstract

Dietary exposure of Singapore population to foodborne and natural toxins was estimated through Total Diet Study (TDS) approach. Among the common mycotoxins and plant toxins studied, such as aflatoxins, ochratoxin A, zearalenone, deoxynivalenol, and fumonisins, aflatoxins were identified with food safety concerns. Aflatoxin occurrence was determined in 642 commonly consumed foods, with a detection rate of 4%, and a mean concentration of 0.01–0.07 µg/kg. Dietary exposure and risk assessment of aflatoxins for the general population revealed a mean estimated daily intake (EDI) of 0.0002–0.002 ng/kg bw/day, a margin of exposure (MOE) of 2819–7101, cancer risk of 0.002–0.004 additional cases per 100,000 person per year, and a hazard quotient (HQ) of 0.19–0.20. Despite the low overall estimated exposure risk for the general population, elevated exposure was observed among the eaters-only group, with the highest upper-bound (UB) exposure reaching 3.4 ng/kg bw/day for high consumers (95th percentile) of satay sauce, a popular Asian delicacy. The corresponding cancer risk of 0.23 additional cases per 100,000 individuals, or 14 additional cases annually, contributes to an estimation of 1% of the 1442 liver cancer cases reported in Singapore in 2022. These findings highlight the importance of continuous monitoring and call for appropriate mitigation strategies for further reduction in aflatoxin exposure in the Singapore population.

## 1. Introduction

Aflatoxins (AFs), a group of naturally occurring carcinogenic compounds, are toxic secondary metabolites produced by microscopic fungi, *Aspergillus flavus* and *Aspergillus parasiticus* [1]. AFs are one of the most prevalent mycotoxins, other than ochratoxin A, fumonisins, zearalenone, type A trichothecenes (e.g., T2–toxin), type B trichothecenes (e.g., deoxynivalenol), and patulin [2,3,4]. Among the twenty known AFs, the four main AFs are aflatoxin B1 (AFB1), aflatoxin B2 (AFB2), aflatoxin G1 (AFG1), and aflatoxin G2 (AFG2), collectively referred to as Total Aflatoxins (AFT) in this manuscript, with order of toxicity AFB1 > AFG1 > AFB2 > AFG2 [5]. AFs are classified by the International Agency for Research on Cancer (IARC) as Group 1 carcinogen, causing hepatocellular carcinoma (HCCs) [6,7], with AFB1 as the strongest naturally occurring genotoxic carcinogen, targeting the liver. Due to its carcinogenic nature, exposure to AFs through food consumption should be kept as low as possible. Chronic exposure to AFB1 is known to increase liver cancer risk when associated with hepatitis B virus infection [8]. Chronic intake of AFs, coupled with malnutrition, may also result in immunosuppression, impaired growth, and various diseases [9].

Climate change has altered the distribution, appearance, and resilience of the fungal species and the production of mycotoxins [10,11,12]. Crops can also be affected by the abiotic stress factors, rendering them more susceptible to fungal infections and mycotoxins contamination [11]. Among these mycotoxins, AFs pose a significant public health concern due to their widespread presence in food and feed commodities [13,14]. Crops critical to global food security, including grains, nuts, maize, cereals, spices, dried fruits, crude vegetable oils, seeds, cocoa beans, and wine, are particularly vulnerable to fungal contamination before and after harvest [15,16]. Protecting these high-risk crops from the impact of climate change is a pressing priority, demanding research advancement, food production technology innovation, and food safety risk management. 

AFs were evaluated in the 31st meeting (1987) of the Joint FAO/WHO Expert Committee on Food Additives (JECFA) as potential human carcinogens [17]. Following the conclusion by IARC for AFs as carcinogens in 1993 [18], JECFA recommended, during the 46th meeting (1997), that the exposure to AFs should be reduced to as low as reasonably practicable [19]. Taking the JECFA evaluations as reference, Codex Alimentarius Commission (CAC) set the maximum levels (MLs) of 10–20 μg/kg AFT for a range of food commodities, with 10 μg/kg for cereal-based food for infant and young children, aiming to reduce AFs levels in food supply [20]. Singapore Food Agency (SFA) set MLs at 5 μg/kg for either AFT or AFB1 in food, and 0.1 μg/kg for AFB1 in infant food, under the Singapore Food Regulation (SFR).

Throughout recent years, the testing for foodborne and natural toxins has evolved along with sophisticated technological advancement, aside from the conventional analytical method such as high-performance liquid chromatography (HPLC) and thin-layer chromatography (TLC). The growing demand for rapid detection has led to the development of rapid and sensitive analytical methods using novel enzyme-linked immunosorbent assay (ELISA) [21], biosensors [22,23,24], microfluidic immunoassay [24], test strip [25], and spectroscopy [26], as well as multi-components analysis using liquid chromatography tandem mass spectrometry (LC-MS/MS) and liquid chromatography high-resolution mass spectrometry (LC-HRMS) [27].

Assessment of dietary exposure to food chemicals through Total Diet Study (TDS) approach has been widely applied, due to its representativeness of populations’ dietary habits and the analysis of food in normal consumption conditions [28,29]. This approach was adopted across countries, e.g., United Kingdom, France, Netherland, Ireland, Lebanon, Australia, New Zealand, Africa, China, Hong Kong, Malaysia, and Vietnam in the assessment of dietary exposure to mycotoxins [30,31,32,33,34,35,36,37,38,39,40,41,42,43,44,45]. 

This study investigated the presence of mycotoxins and plant toxins in 501 samples of commonly consumed foods collected through TDS [46]. The toxins analyzed included potential carcinogens (AFT, aflatoxins M1 and M2, ochratoxin A, fumonisins B1 and B2, and pyrrolizidine alkaloids), and non-carcinogens (zearalenone, deoxynivalenol, ergot alkaloids, and tropane alkaloids). A preliminary dietary risk assessment was conducted for the general Singapore population, using mean consumption data for the entire population mean consumers. The calculated estimated daily intake (EDI), margin of exposure (MOE), cancer risk, and hazard quotient (HQ) values indicated negligible acute and chronic health risks for most toxins. However, slight concerns were identified for AFs, with 2% of the upper bound (UB) MOEs falling below the safety margin of 10,000 for carcinogenic compounds for the entire population. Consequently, an additional 141 food samples were collected and analyzed specifically for aflatoxin occurrence, bringing the total sample size to 642.

In-depth dietary exposure assessments for AFs revealed elevated liver cancer risk for high consumers of several popular local foods. The implicated foods, in decreasing order of concern, are satay sauce, soy milk, and kueh. These insights are valuable for potential post-study actions, such as formulating consumer advisories, reviewing regulatory standards, and strengthening food safety mitigation strategies, to protect public health from the harmful health effects of AFs exposure. 

To the best of our knowledge, this is the first study on dietary AF exposure with trend analysis in Singapore. 

## 2. Results

### 2.1. Occurrence of Mycotoxins and Plant Toxins in Commonly Consumed Food in Singapore

The 501 food samples from the TDS exhibited generally low detection rates for mycotoxins and plant toxins, ranging from 0.2 to 3%. The determined concentration ranges and corresponding detection rates are 3.87–50.9 µg/kg (3%) for deoxynivalenol, 0.11–12.62 µg/kg (2%) for ochratoxin A, 1.23–68.78 µg/kg (1%) for ergot alkaloids, 5.74–65.4 µg/kg (1%) for fumonisins, 5.6–10.3 µg/kg (1%) for zearalenone, single detection of 22.45 µg/kg (0.2%) for pyrrolizidines alkaloids, and no detection for aflatoxin M1, M2, and tropane alkaloids.

In short, mycotoxins and plant toxins were detected at low prevalence and concentrations, with some compounds, including aflatoxin M1, M2, and tropane alkaloids, remained undetected.

### 2.2. Occurrence of AFs (as AFT) in Commonly Consumed Foods in Singapore

The food samples collected and prepared under TDS were categorized into nine food categories for a better representation of food products associated with AF occurrence and subsequent exposure risk assessment. These nine food categories comprise nuts and seeds, grains products, rice products, infant food, bakery and confectionary, other plant products, animal products, sauces and condiments, and beverages. Mean AFT concentrations were calculated for each food category (Table 1, refer to Appendix A for sub-categories expansion), expressed as medium bound (MB; ND = LOD/2) and concentration range of lower bound (LB; ND = 0) to upper bound (UB; ND = LOD). 

AFs were detected in 25 of the 642 food products (4%), with a mean concentration ranging from 0.01 to 0.07 µg/kg. The top three food categories with mean total aflatoxin concentrations in descending order, are nuts and seeds (0.27–0.30 µg/kg), sauces and condiments (0.09–0.14 µg/kg), and bakery and confectionary (0.05–0.11 µg/kg). 

In summary, AFs were found in 4% of foods tested, mainly in nuts, sauces, and baked goods, with nuts having the highest levels.

### 2.3. Estimated Daily Intake (EDI)

Mean and 95th percentile exposure levels were used to represent the exposure of average and high consumers in two main groups of consumers; the entire population (including those who consume and those who do not consume the specific food products) and the eaters-only (those who consume the specific food products), resulting in four consumer groups: entire population mean consumers, entire population high consumers, eaters-only mean consumers, and eaters-only high consumers. These four groups were carried forward into the risk assessment part.

The EDI was tabulated for each food category (Table 2 presents data for the entire population mean consumers and eaters-only high consumers; Appendix A includes all four consumer groups with expanded sub-categories). The mean EDI for the entire population mean consumers is 0.0002–0.002 ng/kg bw/day (Table 2).

In summary, AF exposure was evaluated for both average and high consumers within the entire population and eaters-only groups, revealing a very low average intake of 0.0002–0.002 ng/kg bw/day for the general population.

### 2.4. Margin of Exposure (MOE)

The MOE was calculated using the EDI with the reference point recommended by the JECFA [47] and European Food Safety Authority (EFSA) [48] (BMDL_10_ = 0.17 µg/kg bw/day). MOEs to AFs for Singaporeans are presented in a range (LB–UB) for each food category, using the lowest figures for LB (ND = 0) and UB (ND = LOD), respectively (Table 2 presents data for the entire population mean consumers and eaters-only high consumers; Appendix A includes all four consumer groups with expanded sub-categories). The MOE showed a decreasing trend from the entire population mean consumers to high consumers, and from eaters-only mean consumers to high consumers.

The exposure assessment for mean consumer across the entire population indicates low concern, with LB MOEs generally exceeding 10,000, suggested low exposure risk to public health for AFs as genotoxic and carcinogenic substances [49]. An exception was noticed in the legumes sub-category (7101) under “other plant products”, where LB MOEs fall far below 10,000, suggesting potential cancer risk with chronic intake. For UB MOEs, several food categories showed elevated cancer risk concerns, with MOEs falling below 10,000, including infant food (2819), beverages (3891), rice products (4541), animal products (6126), other plant products (legumes, 6519), and grains (8367).

For high consumers (95th percentile) in the eaters-only group, the UB MOEs for all food categories fall below the safety margin of 10,000, ranging from 51 to 725 (lowest figure for the worst-case representation). The UB MOEs stretched to as low as 51 for sauces and condiments, 80 for legumes under other plant products, and 122 for bakery and confectionery. This indicates potential health concern for high consumers of these food products and highlights the need for closer scrutiny and possible intervention.

In short, exposure assessments indicate minimal risk for the general population; however, elevated risks associated with specific food categories such as legumes, infant foods, and sauces among high consumers warrant close monitoring and mitigation. 

### 2.5. Cancer Potency (Risk)

The cancer risks from exposure to AFs in the Singapore population are calculated for the four consumer groups (Table 2 presents data the for entire population mean consumers and eaters-only high consumers; Appendix A includes all four consumer groups with expanded sub-categories), expressed as cancer cases per 100,000 persons per year. Similarly, cancer risk is presented in a range, LB (ND = 0) to UB (ND = LOD), taking the highest value for the worst-case simulation.

For mean consumers in the general population, the UB cancer risk is estimated to be 0.004 additional cancer cases per 100,000 person per year, which is lower than the guidance value of 0.014 [50]. Based on Singapore’s population of 6.036860 million (from the Department of Statistics Singapore 2024 data [51]), the exposure to AFs through food consumption might lead to 0.24 additional cancer cases per year in Singapore. This contribution (0.02%) is negligible when compared with 1442 liver cancer cases in Singapore in 2022 [52]. 

For high consumers in the eaters-only group, the highest UB cancer risk is estimated to be 0.23 additional cases per 100,000 person per year (Table 3, entry 1), attributed to the sauces and condiments category. The implicated food product is satay sauce, with a total aflatoxin concentration of 0.83 µg/kg and an EDI of 3.4 ng/kg bw/day, based on consumption data of 3.894 g/day for high consumers (95th percentile) of eaters-only group. This contributes to an estimation of 14 (1%) additional liver cancer cases annually, based on Singapore’s population of 6.036860 million (from Department of Statistics Singapore 2024 data [51]) and 1442 liver cancer cases in 2022 [52]. 

The second and third highest UB cancer risk of 0.15 and 0.10 additional cases per 100,000 person per year come from soy milk (legumes sub-category) and kueh (bakery and confectionery category), a type of snack made with glutinous rice and coconut. With respective EDIs of 2.1 and 1.4 ng/kg bw, these categories have an estimated contribution of about nine (0.6%) and six (0.4%) additional liver cancer cases in Singapore yearly, respectively (Table 3).

In general, the cancer risk from AF exposure in Singapore remains low for the general population. Nonetheless, higher risks are observed among high consumers of satay sauce, soy milk, and kueh, with a potential increase of 6 to 14 cases per year.

### 2.6. Hazard Quotient (HQ)

The HQ for immunotoxicology effects from AF exposure via food consumption was assessed in the Singapore population. A worst-case scenario was considered by assessing the exposure for the high consumers in the eaters-only group. This exposure was compared against the lower value (0.017) of the reference range, 0.017 to 0.082 µg/kg bw/day [53]. Additionally, the highest values for each food category were presented in a range, from LB (ND = 0) to UB (ND = LOD). All the values were less than 1, indicating low safety concern in this aspect (Appendix A). 

In summary, the potential immunotoxic effects of AF exposure within the Singaporean population revealed low safety concern even under worst-case scenarios.

## 3. Discussion

### 3.1. High-Risk Consumers Versus High-Risk Food Products

While this study indicates a low overall cancer risk from AF exposure in Singapore general population, certain food products pose a greater threat to high consumers, particularly for high-risk food products such as nuts, seeds, legumes, and bakery products. These findings are aligned with established knowledge, as food products in these food categories are well-known for their susceptibility to AF contamination and are ranked highly among the most contaminated crops. Peanuts, a popular snack, with especially high consumption during Chinese New Year, are a known source of AFs. While often promoted as healthy snack for pregnant women due to their exceptionally high nutritional value (e.g., high in protein, healthy fats, fiber, folate, and iron content), their potential AF contamination should not be under-estimated. This concern extends to satay sauce, a popular local condiment, made primarily from ground roasted peanuts, often consumed with satay and satay bee hoon. Soy milk, in the high-risk legume sub-category, is widely consumed for its purported health benefits (e.g., lowering blood pressure, cholesterol, and weight). Kueh, a traditional Southeast Asian snack made from rice or glutinous rice (another susceptible crop), is also widely consumed, with increased consumption during Hari Raya Puasa. Our study indicates that liver cancer risk for the high consumers of these popular local delicacies will be elevated due to higher AFs exposure. 

Based on our data from the national surveillance program for AFs, there were occasional spikes in AFs occurrence in certain food products, e.g., peanuts, peanut butter, and maize flour. Although the assessment of acute toxicity for isolated cases was low, the long-term cumulative carcinogenicity effect could not be ignored, and this calls for stringent mitigation measures laid down in later discussion. 

### 3.2. Comparison with International Total Diet Studies on Aflatoxin Exposure 

AF exposure in Singapore was compared to that of Malaysia and Hong Kong, countries with similar dietary patterns (Table 4). Singapore exhibited the lowest exposure, likely due to its smaller UB mean aflatoxin concentration (0.07 µg/kg) as compared to Hong Kong (1.42 µg/kg) and Malaysia (1.67 µg/kg). This difference may also be partly attributed to the variations in handling of left-censored data, as both Hong Kong and Malaysia used slightly higher limits of detection (LOD) and quantification (LOQ). While both Singapore and Hong Kong showed the highest AFs concentrations in legumes, nuts, and seeds (nuts and seeds: 0.27–0.30 µg/kg and legumes: 0.13–0.19 µg/kg in Singapore; peanut butter: 6.34–6.37 µg/kg and peanuts: 1.64–1.79 µg/kg in Hong Kong), the exposure ranges differed significantly. 

Comparison of dietary exposure (mean EDI) to AFs with other countries is summarized in Table 4. There is a large variation in the estimated daily intake (EDI) reported by different countries, which likely stems from the differences in left-censored data treatment (LB/UB with respect to LOD/LOQ), analytical methods, food consumption data collection methods, dietary habits, and study periods. Globally, a trend of lower EDI is observed in developed countries (e.g., Japan, UK, France, The Netherland) as compared to developing countries (e.g., Vietnam, China, Africa). This is likely due to a combination of advanced post-harvest processing technologies, better storage facilities, and stricter regulatory limits in developed countries, leading to overall lower AF contamination levels in the staple food. 

The contributing food products with high AF concentration vary for different countries (Table 4–major source of AFs contributor). It is noteworthy that AFs exposure is inevitable, given that the main source of exposure is through staple foods, such as rice, bread, flour, and cereal products. Regardless of the dissimilarities in dietary preference for staple food across nations from different geographical regions, AFs exposure through staple food is a risk factor that warrants attention.

### 3.3. Potential Mitigation Measures for Reduction in Dietary AF Exposure

AFs contamination of crops is a global concern due to its hepatotoxicity and prevalence throughout the food supply chain—from plantation, harvesting, to post-harvest treatment (e.g., drying, storage, transit and distribution). For Singapore, as a country relying heavily on imported food, minimizing contamination during the initial stages (plantation, harvesting, post-harvest treatment) could be achieved through audit and accreditation of food sources, with enforcement of good agricultural practices (GAPs) and good manufacturing practices (GMP). For the latter stages (storage, transit and distribution) which are more relevant in Singapore context, regular audits by the government authority in local facilities is important to ensure compliance to the food safety regulation.

Singapore employs an integrated sampling plan that monitors both imported raw food and retailed processed products. This system ensures continuous surveillance and efficient screening of food products associated with potentially high AFs levels. It enables timely triggering of downstream risk control actions, such as food recalls or suspension of imports from problem sources, when the detection of AFs exceeds the ML or when the MOE falls below 10,000. Through this on-going integrated system, the ultimate goals of minimizing exposure to these carcinogens to protect public health could be achieved. The approach extends to other mycotoxins and plant toxins as well. 

Enhancing consumer awareness through education is another important risk control measure not to be overlooked, targeting especially to the high consumers of the identified high-risk food products (e.g., satay sauce, soy milk, and kueh). For example, dedicated community outreach programs, road shows, workshops, healthy lifestyle programs, and school programs could provide targeted education to the vulnerable consumers, considering the different diet cohorts, such as the elderly for kueh consumers; pregnant women, children and their parents, health-conscious groups for soy milk consumers; and satay sauce consumers across ethnics. Due to the current trend of digital world, risk-at-a-glance articles at SFA’s website and social media platforms, such as Instagram and Tik Tok, would be preferred for young consumers. In addition, consumer education on high-risk food categories and increased awareness on moldy foods (not to consume the non-moldy part); as well as promoting dietary diversification is crucial to minimize AF dietary exposure. 

Furthermore, following the findings on the high exposure risk to AFs stemming from the high consumption of satay sauce, as well as soy milk and kueh, reaching out to the one-health platform and health care institution groups such as Singapore Health Services (SingHealth), would be a prioritized follow-up measure to allow larger scale of consumer engagement for better enhancement of the effectiveness of public education.

### 3.4. Strengths and Limitations

This study represents the first comprehensive assessment of AF occurrence in commonly consumed foods in Singapore. The detailed exposure assessment for AFs was carried out for each food product, based on its respective consumption data, and followed through the MOE, liver cancer risk, and HQ; instead of having single MOE, cancer risk and HQ values for each food category based on the mean concentrations (or EDI). This approach provides a more nuanced understanding of dietary exposure and risk by revealing potential high-end risks through minimizing the masking effect of averaging. In addition, the risk evaluation part was based on the worst-case scenario, through the highest figures for cancer risk and HQ, and lowest figures for MOE in each food category, enabling the tracing of the extreme risk to the responsible food product.

A deterministic approach was used in the risk assessment in this study due to its wide use in chemical risk assessment, its simplicity and ease of application, and partly due to the lack of probabilistic modeling software. However, this approach gave rise to limitation of over-estimation of the risk presented, by using single-point estimates with the assumption that the same average amount of specific food products was consumed by the specific consumer groups, and that the AF concentration is always present at the fixed concentration detected. 

While the literature on AF exposure and risk assessments often uses AFT or AFB1 individually, or in combination, AFT was used in this study. Although this may lead to a slight over-estimation of liver cancer risk, it allows for a more conservative estimation of risk, ensuring a lower level of AF exposure in the Singaporean population. Furthermore, part of the over-estimation of liver cancer risk stems from the applied UB by summing the LODs for AFB1, AFB2, AFG1, and AFG2.

The literature has shown that AFs content could be reduced by thermal cooking treatment [14,56,57,58,59], even with simple cooking method achieving a reduction of 34% AFB1 content in rice [59]. In view of this, future research could explore the impact of food processing techniques, such as different cooking methods, on final product’s AF content, an area not addressed in this study.

## 4. Conclusions

This study provides the first estimation of dietary AF exposure for the Singaporean population using a Total Diet Study (TDS) approach. While average consumers are generally exposed to low AF contamination levels, a potential exposure risk is identified for high consumers of certain popular local food products that are associated with higher AF contamination (e.g., satay sauce, soy milk, and kueh). The highest estimated UB liver cancer risk associated with these foods is up to 14 (1%) of additional annual cases. Therefore, continuous food product monitoring in Singapore, along with regular audits and inspections of storage facilities, is crucial to further reduce AF exposure and protect public health. Targeted consumer education, particularly for high consumers and vulnerable groups, is also essential.

## 5. Materials and Methods

### 5.1. Chemicals and Reagents

Easi-Extract Aflatoxin Immunoaffinity Column (IAC) (RP70N) and Kobra Cell (RBRK01) were purchased from R-Biopharm (Darmstadt, Germany). HPLC grade acetonitrile was purchased from Tedia (Fairfield, OH, USA), while HPLC grade methanol from Elite Advanced Materials Sdn Bhd (Rawang, Malaysia). Ultra-pure water was produced by ELGA PureLab Ultra Analytic ultra-pure polishing system from Veolia (High Wycombe, UK), with a resistance of 18.2 MΩ-cm. Analytical grade nitric acid was from Fisher Scientific (Leicester, UK), potassium bromide was from Sigma-Aldrich (St. Louis, MO, USA), sodium chloride was from Schedelco (Singapore), and Dulbecco A phosphate-buffered saline tablets was from Oxoid Limited (Basingstoke, UK) were used.

### 5.2. Standard Solutions

Aflatoxins B and G mixture (Biopure) was purchased from Romer Labs (Tulln, Austria). 100 ng/mL AFB1/G1 (25 ng/mL AFB2/G2) was prepared from the stock solution of 2 µg/mL AFB1/G1 (0.5 µg/mL AFB2/G2). The calibration standards were prepared through serial dilution in Acetonitrile: Water (1:1), with AFB1/G1 concentration ranges of 0.3125–10 ng/mL.

### 5.3. Sample Collection and Preparation

The Singapore TDS was carried out between 2021 and 2023 [46], encompassing foods commonly consumed by the Singapore population, with the selection of food based on the Food Consumption Survey through 24 h recall interview by an external market research company engaged by SFA. 

An initial categorization included 24 food categories, which were further merged and re-categorized into 9 food categories based on AF susceptibility, comprising nuts and seeds, grains products, rice products, infant food, bakery and confectionary, other plant products (e.g., fruits, vegetables, legumes, roots and tubers, etc.), animal products, sauces and condiments, and beverages, for subsequent dietary exposure and risk assessment of AFs.

### 5.4. Method Performance

The High-Performance Liquid Chromatography with Fluorescence Detector (HPLC-FLD) method was validated at 0.05, 0.1 and 0.2 μg/kg of AFB1/G1 (0.0125, 0.025 and 0.05 μg/kg of AFB2/G2) in infant food samples with spiked recovery of 70–120%. Triplicate analyses over 2 occasions were conducted for repeatability and intermediate precision determination. The relative standard deviations (RSD) were less than 20%. The limit of detection (LOD) and limit of quantitation (LOQ) were determined to be 0.025 μg/kg AFB1/G1 (0.00625 μg/kg AFB2/G2), and 0.05 μg/kg AFB1/G1 (0.0125 μg/kg AFB2/G2), respectively, across 7 spiking replicates. 

### 5.5. Preparation and Treatment of Samples for AFs Analysis

25 ± 0.2 g of homogenized sample was extracted through blending with 125 mL of Methanol: Water (80:20), followed by filtration and 3-fold dilution with ultrapure water. 60 mL of the diluted filtrate were cleaned up using the IAC. The subsequent 2 mL Methanol eluate was evaporated under nitrogen gas and reconstituted with 250 µL of Methanol: Water (1:1).

### 5.6. HPLC Analysis

HPLC-FLD analysis was performed on an Agilent 1200 HPLC, with post column derivatization using Kobra Cell. Liquid chromatographic separation was achieved on a Shim-pack VP-ODS column (4.6 × 150 mm, 4.6 µm, Shimadzu, Japan), with injection volume of 25 µL and a flow rate of 1 mL/min for 12.50 min in an isocratic run, using mobile phase of methanol–water (45:55) with 119 mg of potassium bromide and 87.5 µL of concentrated nitric acid. The excitation and emission wavelength were 350 nm and 450 nm, respectively.

### 5.7. Dietary Exposure of AFs 

#### 5.7.1. Data Analysis

The data was analyzed using Excel with Microsoft 365. Excel was used for consumption data matching to individual food products, left-censored data treatment, EDI, and risk assessment components (MOE, cancer risk, HQ) calculation. Mean values of concentration and EDI were calculated for each food category.

#### 5.7.2. Food Consumption Data

The food consumption data for the Singapore population were obtained from 24-h dietary recall interviews with 2014 participants aged 15 to 92 years in 2 non-consecutive occasions during the period 2021–2022. The mean consumption amount for each food product was averaged from the consumption amount reported by individual respondents. Consumption amounts were calculated for the entire population, in terms of mean, median, and 95th percentile, normalized with 2014 respondents. Another set of consumption amounts was calculated for eaters-only, also in terms of mean, median, and 95th percentile, with normalization by the no. of eaters (ranging from 1 to 2310) for the respective food product, contributing to relatively higher consumption data in comparison with the entire population.

#### 5.7.3. Left-Censored Data Treatment

With a detection rate of 4%, 96% of the results fall below the LOD of 0.025 μg/kg AFB1/G1 (0.0125 μg/kg AFB2/G2) [28,60], and are reported as not detected (ND). These are referred to as left-censored data, where the values could be anywhere between zero and LOD, and the underlying uncertainties could be introduced into the subsequent exposure assessment. The deletion of these data was not recommended, due to the resultant over-estimation of the associated risk. For food risk assessment, the substitution method is recommended for treating left-censored data [60]. Following guidelines from WHO GEMS/Food-EURO [61] and EFSA/FAO/WHO [28,60], we employed this method using the LB and UB approach as the default standard. Two estimates were calculated: the LB data, where ND values were substituted with zero (ND = 0), and the UB data, where ND values were substituted with LOD (ND = LOD). 

Despite the drawbacks of certain biases and non-consideration of the overall distribution of positive samples, this method is widely used due to its ease of implementation and the moderation of conservative estimation of the exposure in relation to the over-estimation of mean and under-estimation of the variability.

#### 5.7.4. Statistical Analysis of AF Concentration

For each food category, the mean concentration of all the food products is presented in a range of LB (ND = 0) to UB (ND = LOD). This format was followed through into dietary exposure and subsequent risk assessment. For simplified presentation, medium bound (MD) data of AFT estimated from ND = LOD/2 were included.

#### 5.7.5. Dietary Exposure

The dietary exposure to AFs was determined using Total Aflatoxins (AFT)—the summation of AFB1, AFB2, AFG1, and AFG2—instead of AFB1 (Group 1 carcinogen).

Although a probabilistic approach is known to obtain the best estimate of dietary exposure [62], in many cases, a deterministic approach as a screening method is appropriate to ascertain the risk [36,37,63,64]. In this study, a deterministic approach was used to estimate the dietary exposure to AFs (ng/kg bw/day) in an individual consumer. Food consumption data was applied to each food product, with normalization of the consumption data for multiple varieties of same food product (e.g., different types of buns). The subsequent EDIs for each food product were calculated using the corresponding occurrence data. The dietary exposure to AFT was estimated by multiplying the concentration level (ng/g) for a certain food product by the consumption data of the same food product (g/kg bw), where the daily intake (g/day) was normalized by the body weight (kg bw) of the participants.

The equation is as followsEDI (ng/kg bw/day) = C (ng/g) × DI (g/day)/Body Weight (kg bw)(1)
where EDI is the estimated daily intake, C is the concentration, and DI is the daily intake of each food product, with normalization if required [65]. 

The mean EDI for every food category was presented in a range, with the mean LB formulated from ND = 0, and mean UB formulated from ND = LOD. These mean EDIs were used to compute the mean EDI for all the 642 samples. 

### 5.8. Risk Assessment and Characterization of AFs 

The risk characterization of the carcinogenicity effect of long-term exposure to AFs was estimated using two approaches, MOE, and cancer potency. The estimation of MOE from the dietary exposure data was based on the BMDL_10_ of 0.17 µg/kg bw/day, adopted by both JEFFA (2018) [47] and EFSA (2007) [48] for the MOE approach [47,48,66]. References from JECFA (2018) [47] and EFSA (2020) [65] were used in the cancer potency estimations. 

Besides carcinogenicity, exposure to AFs was shown to exert immunological effects (non-carcinogenic) in relation to innate and adaptive immunity [53]. An additional risk assessment tool, HQ, was therefore applied, based on the Tolerable Daily Intake (TDI) for aflatoxin-related immune impairment in the range of 0.017 to 0.082 µg/kg bw/day [53], using the lower value of 0.017 to simulate the worst-case estimation.

In this study, the calculated EDIs were compared with the adopted reference values for the subsequent formulation of MOE, liver cancer potency (cancer risk) and HQ, presented in a range of LB–UB for each food category. To simulate the worst-case scenario, the lowest values were used for MOE, the highest values were used for cancer risk and HQ; cancer risk was based on 95% upper confidence bound potency estimates, and HQ was calculated for eaters-only high consumers. 

#### 5.8.1. Margin of Exposure (MOE)

Benchmark dose (BMD) is used as the point of departure (POD) for deriving human health-based guidance values. Benchmark Dose Lower Confidence Limit (BMDL) is the lower one-sided confidence limit of the benchmark dose (BMD) for benchmark response (BMR) in bioassay for cancer in animals. For the assessment of MOE to AFs, BMDL_10_ of 0.17 µg/kg bw/day recommended by JECFA [47] and EFSA [48] (as the lower limit on the BMD for a 10% response) was adopted. An MOE of 10,000 or higher indicates low risk for genotoxic and carcinogenic substances from public health point of view, as stated by the EFSA Scientific Committee [49]. 

The MOE was obtained from the following equation MOE = BMDL_10_ (µg/kg bw/day)/EDI (µg/kg bw/day)(2)
where BMDL_10_ is the benchmark dose lower confidence limit, and EDI is the estimated daily intake.

The MOE values are presented in a range, LB (ND = 0) to UB (ND = LOD), taking the lowest MOE values for worst-case simulation.

#### 5.8.2. Cancer Potency

Positive correlation between AFB1 exposure and primary liver cancer (hepatocellular carcinoma) was shown in most epidemiological studies, with enhanced risk for simultaneous exposure to hepatitis B virus (HBV) [8,66,67]. Hepatocellular carcinoma (HCC) is a leading cause of cancer mortality in Asia, as the 6th most common cancer in the world, and the 3rd most common cause of cancer deaths globally [68]. 

In Singapore, HCC is the 5th most common cancer affecting men [69]. A total of 1442 cases of liver cancer was reported in 2022 in Singapore [52], which is 24 cases per 100,000 persons per year, based on Singapore population of 6.036860 million from Department of Statistics Singapore 2024 data [51].

Cancer potencies estimation recommended by JECFA [47] was used in the estimation of cancer risk, incorporating the incident of individuals with HBsAg+ (HBV) and the carcinogenic potency of AFs. The incidence of liver cancer (Pcancer) from exposure to AFB1 was estimated by the following equation Pcancer = [(PHBV+) × (HBV+) + (PHBV−) × (HBV−)] × (AFexposure)(3)
where Pcancer is the cancer risk.

PHBV+ is the potency estimates for the HBV+ fraction of the population (prevalence of HBV carrier).PHBV− is the potency estimates for the HBV fraction of the population (prevalence of non-HBV carrier).HBV+ is the population fraction of chronic HBV cases (individuals with HBV or serum hepatitis B surface antigen-positive (HBsAg+)).HBV− is the population fraction of non-chronic HBV cases (individuals without HBV or serum hepatitis B surface antigen-negative (HBsAg−)).

The prevalence of HBV+ in Singapore is 4.09% [70]. The estimated cancer potency for exposure to 1 ng AFB1/kg bw/day is 0.017 (0.049 as 95% UB) for HBV+ population, and 0.269 (0.562 as 95% UB) for HBV- population, in terms of additional cancer cases per 100,000 person per year [47]. By substituting into Equation (3), Pcancer was obtained at 0.027 (0.070 as 95% UB) aflatoxin-induced cancer cases per 100,000 person per year for exposure to 1 ng AFB1/kg bw/day. 

According to the World Health Organization (WHO) Guideline for drinking-water quality [50], an excess lifetime cancer risk of 10^−5^ or less is considered to be of low risk for health concern for genotoxic carcinogens. By assuming a lifetime exposure of 70 years, the value 10^−5^ corresponds to a yearly excess cancer risk of 0.014 additional cancer cases per 100,000 persons or 0.14 cases in 1 million persons after lifetime exposure of AFB1, which was used as the cut-off criteria for cancer risk.

#### 5.8.3. Hazard Quotient (HQ)

HQ is the ratio of the potential exposure to the substance to the level at which no adverse effect is expected, such as TDI or acute reference dose (ARfD) [71,72,73]. The equation used is as follows:HQ = EDI (µg/kg bw/day)/TDI or ARfD (µg/kg bw/day)(4)

The reference value based on TDI for aflatoxin-related immune impairment for calculation of HQ were 0.017–0.082 μg/kg bw/day [53]. The lower value of 0.017 μg/kg bw/day was used for the worst-case estimation. HQ value of greater than 1 indicates non-tolerable exposure. 

## Figures and Tables

**Table 1 toxins-17-00324-t001:** Summary of AFT concentrations in 9 food categories.

Entry	Food Category	n ^a^	n+ ^b^	Mean AFT ^c^ Concentration (µg/kg)	
				MB ^d^	LB–UB ^e^
1	Nuts and seeds	4	3	0.28	0.27–0.30
2	Grains products	26	2	0.04	0.01–0.07
3	Rice products	17	2	0.04	0.01–0.07
4	Infant food	146	6	0.03	0.003–0.06
5	Bakery and confectionary	18	1	0.08	0.05–0.11
6	Other plant products (i.e., legumes)	212	4	0.04	0.01–0.07
7	Animal products	175	0	0.03	0–0.06
8	Sauces and condiments	28	5	0.12	0.09–0.14
9	Beverages	16	2	0.05	0.02–0.08
Total		642	25	26.55	7.81–45.49
Mean		–	–	0.04	0.01–0.07

^a^ n: number of composite samples. ^b^ n^+^: number of positive samples. ^c^ Total Aflatoxins (AFT): sum of AFB1, AFB2, AFG1 and AFG2. ^d^ Medium bound (MD) denotes ND = LOD/2. ^e^ Lower bound (LB) and upper bound (UB) denote ND = 0 and ND = LOD, respectively.

**Table 2 toxins-17-00324-t002:** Summary of mean EDI, MOE, and cancer risk for 9 food categories.

Entry	Food Category	Mean EDI (ng/kg bw/day) LB–UB ^a^		MOE LB–UB ^a^		Cancer Risk (Cases/100,000 Persons/year) LB–UB ^a^	
		Entire Population Mean Consumers	Eaters-Only High Consumers	Entire Population Mean Consumers	Eaters-Only High Consumers	Entire Population Mean Consumers	Eaters-Only High Consumers
1	Nuts and seeds	0.001–0.001	0.108–0.125	47,619–44,321	779–725	0.0002–0.0003	0.015–0.016
2	Grains products	0.001–0.003	0.026–0.196	12,850–8367	415–270	0.001–0.001	0.029–0.044
3	Rice products	0.001–0.012	0.028–0.260	17,329–4541	557–393	0.001–0.003	0.021–0.030
4	Infant food	0.00001–0.001	0.002–0.040	258,097–2819	1290–168	0.00005–0.004	0.009–0.071
5	Bakery and confectionary	0.0001–0.001	0.078–0.179	70,833–12,891	122–122	0.0002–0.001	0.098–0.098
6	Other plant products (i.e., legumes)	0.0001–0.002	0.013–0.128	7101–6519	87–80	0.002–0.002	0.137–0.149
7	Animal products	0–0.001	0–0.130	*–6126	*–166	0–0.002	0–0.072
8	Sauces and condiments	0.001–0.002	0.162–0.238	15,755–15,183	53–51	0.001–0.001	0.226–0.235
9	Beverages	0.0002–0.007	0.091–0.404	110,390–3891	171–135	0.0001–0.003	0.070–0.088
	Total	0.103–1.217	12.028–82.092	-	-	-	-
	Mean	0.0002–0.002	0.019–0.128	-	-	-	-
	Relative Standard Deviation (RSD)	8.93–2.52	9.29–1.66				
	Worst-case	-	-	7101–2819	53–51	0.002–0.004	0.226–0.235
				(lowest figures)		(highest figures)	

^a^ Lower bound (LB) and upper bound (UB) denote ND = 0 and ND = LOD, respectively. * MOE could not be calculated due to either zero concentration (ND) or near-zero consumption data, or a small number of eaters-only, resulting in a non-consumer occupying the 95th percentile position. Note: MOE and cancer risk were presented using lowest and highest figures in each food category, respectively, to represent the worst-case scenario.

**Table 3 toxins-17-00324-t003:** High cancer risk food categories for high consumers (95th percentile).

Entry	Food Category	Implicated Food	AFT ^a^ Concentration (µg/kg)	Eaters-Only High Consumers (95th Percentile)				
				Consumption/day (g/day)	UB Values (ND = LOD)			
					EDI (ng/kg bw/day)	MOE	Cancer Risk (Cases/ 100,000 Persons/year)	HQ
1	Sauces and condiments	Satay sauce	0.83	3.894	3.4	51	0.23	0.19
2	Other plant products (i.e., Legumes)	Soy milk	0.42	4.651	2.1	80	0.15	0.12
3	Bakery and confectionary	Kueh	0.96	1.454	1.4	122	0.10	0.08

^a^ Total Aflatoxins (AFT): sum of AFB1, AFB2, AFG1 and AFG2. Note: MOE, cancer risk, and HQ are presented using lowest and highest figures in each food categories, respectively, to represent worst–case scenario.

**Table 4 toxins-17-00324-t004:** Comparison of dietary exposure (EDI) to AFs for the general adult population across different countries.

Country, Year (AF)	Mean Estimated Daily Intake (EDI) (ng/kg bw/day)		Major Sources of AFs Contributor
	Entire Population Mean Consumers	Entire Population High Consumers (95th Percentile)	
Singapore TDS, 2023 (AFT)	0.0002–0.002 (Mean EDI) 0.103–1.217 (Total EDI)	0.001–0.007 (Mean EDI) 0.578–5.926 (total EDI)	Satay sauce, legume, cereal products
Japan market surveillance, 2006 (AFB1) [54]	0.003–0.004	–	Cacao products, peanut butter
Hong Kong TDS, 2013 (AFT) [42,43]	0.2–2.8	0.9–4.9	Cereal products, legumes, nuts and seeds, fats and oil
Malaysia TDS, 2010 (AFT/AFB1) [44] (Scenario-based assessment to exclude data exceeding regulatory limit of 5 µg/kg)	0.47–10.26/0.61–30.09	–	Peanuts
China TDS, 1990 (AFB1) [41]	0.15 (µg/person) 2.3 * (re-computed as ng/kg bw/day)	–	Cereal products
Vietnam TDS, 2016 (AFB1) [45]	35.0–43.7 (total EDI) 3.78–4.34 (re-computed as mean EDI)	–	Rice products
UK TDS, 2014 (AFB1) [30,55]	0–7	0–18	All products were not detected (at LOQ of 0.05 µg/kg)
France TDS, 2007 (AFT) [31]	0.117 (total EDI) 0.030 (re-computed as mean EDI)	0.345 (total EDI) 0.106 (re-computed as mean EDI)	Eggs and egg products (All EDI are relatively small)
France TDS, 2011 (AFT) [32]	0.0019–0.89	0.012–1.54	Bread products, pasta, pastries and cakes
Netherland TDS, 2013 (AFT) [33]	0–1.62	0.033–3.97	Bread and apple
Ireland TDS, 2014 (AFT) [35]	0.23–10.6	0.78–26.9	Cereal products
Lebanon TDS, 2013 (AFB1) [36]	0.63–0.66	1.40–1.46	Bread and toast, nuts, seeds, olives, and dried dates
Lebanon TDS, 2024 (AFB1) [37]	1.26 (total EDI)	–	Traditional food (Kishik, Kibbeh, meat pie)
Africa TDS, 2020 (AFB1) [40] (LB and UB are the lowest and highest mean EDI for different study centers)	4–526	10–1117	Peanut and peanut oil

* Assumption of 65 kg body weight.

## Data Availability

The original contributions presented in this study are included in the article/Appendix A. Further inquiries can be directed to the corresponding author(s).

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
