# Peer review of "First Total Diet Study of Aflatoxins in Singapore: Exposure Risk, High-Risk Foods, and Public Health Implications"

_toxins, 2025, doi:10.3390/toxins17070324_

Round 1

Reviewer 1 Report

Comments and Suggestions for Authors

1. The prevalence of hepatitis B virus (HBV) in Singapore is used to estimate liver cancer risk, yet the source and date of the 6% prevalence figure are not clearly cited or justified. Provide recent and referenced data supporting the HBV prevalence used in cancer risk calculations. If multiple prevalence estimates exist, conduct a sensitivity analysis or clearly justify your selected figure.
2. The manuscript applies lower-bound and upper-bound substitution methods for non-detects but lacks a critical discussion of the limitations and potential bias introduced by these methods. Expand the discussion to address how the use of substitution methods may over- or underestimate exposure and risk. Consider alternative statistical treatments if feasible or justify why the chosen method is sufficient.
3. The definitions and results for the four consumer groups ("whole population mean," "whole population high consumers," "eaters-only mean," and "eaters-only high consumers") are inconsistently labeled across text, tables, and supplementary data. Standardize the terminology and labeling across all tables and figures. Clarify the sampling strategy and statistical rationale for defining “eaters-only” and how they were derived from the dataset.
4. The interpretation of Margin of Exposure (MOE) values as “low” or “high concern” needs further elaboration. For example, the text states that MOE values <10,000 indicate higher concern but does not contextualize what this means for Singapore’s food safety risk management. Provide clearer interpretation of MOE values in relation to JECFA/EFSA thresholds, including what specific public health actions (if any) should be considered when MOEs fall below 10,000.
5. While the discussion mentions surveillance and education strategies, it remains general and lacks specific recommendations based on the study findings (e.g., satay sauce, soy milk, and kueh). Include concrete, evidence-based policy recommendations for high-risk foods and vulnerable populations. Consider whether maximum allowable limits for certain foods (e.g., satay sauce) should be reevaluated.
6. The estimate of 16 additional liver cancer cases annually (1.1% of total cases) for high consumers is a critical finding. However, this risk is not adequately contextualized in terms of public health impact or food regulatory implications. Strengthen the discussion on how this cancer risk should influence national food safety priorities, consumer advisories, or regulatory controls.

Comments on the Quality of English Language

Clarify and standardize acronyms (e.g., AFT, AFs, MOE, HQ) throughout the manuscript and tables. Improve the clarity of key tables (e.g., Table 2, S1–S5) by simplifying formatting and ensuring clear subheadings. Remove placeholder text such as “Author information removal for double blind review” from the final version. Proofread carefully to eliminate typographical and formatting inconsistencies.

Author Response

Thank you so much for the comments and guidance to help us to improve the manuscript.

we have addressed your comments in detail as the attached.

Best Regards

Reviewer 2 Report

Comments and Suggestions for Authors

This manuscript presents the first Total Diet Study (TDS) evaluating dietary exposure to aflatoxins (AFs) in the Singaporean population. A comprehensive risk assessment is conducted, including measurement of AFs in 642 food samples, exposure estimates for various consumer groups, and calculations of health risk indicators such as Estimated Daily Intake (EDI), Margin of Exposure (MOE), cancer risk, and Hazard Quotient (HQ). The findings are contextualized through comparisons with international TDS data and implications for public health policy and consumer education are discussed.

  • Abstract and Title Optimization
  • Suggestion: The current title is clear but could better emphasize the public health impact and novelty. Consider: “First Total Diet Study of Aflatoxins in Singapore: Exposure Risk, High-Risk Foods, and Public Health Implications.”
  • The abstract could briefly mention the percentage contribution of AFs to the total liver cancer burden to enhance impact.
  1. Overreliance on Deterministic Models
  • While a deterministic approach is defensible for screening, its limitations are not sufficiently acknowledged.
  • Introduction should be improved: Please, add examples of the state-of-the-art methods for the detection of  toxins such as Biosensors (doi.org/10.1016/j.microc.2023.108868), HPLC, fluidic (DOI: 10.3389/fchem.2021.626630) etc
    • Suggestion: Add 1–2 sentences in the Methods or Discussion to acknowledge the lack of probabilistic modeling and its implications on exposure uncertainty.
  1. Narrative Structure and Language
  • The manuscript is data-rich but narrative flow is occasionally dense and overly technical for a general scientific audience.
    • Suggestion: Consider summarizing the key takeaways more clearly at the end of each main results subsection.
  • Some language editing is recommended for conciseness and clarity (e.g., “low cancer risk” → “minimal estimated cancer burden”).
  1. Statistical Confidence
  • Confidence intervals or standard deviations are often missing for calculated values (e.g., MOE, EDI).
    • Suggestion: Include at least one set of CI or RSD values in the main text or Supplementary Tables to give a sense of data variability.
  1. Processing Effects Not Addressed
  • The impact of food processing (e.g., cooking, fermentation) on AF content is mentioned as a limitation but could be expanded.
    • Suggestion: Briefly mention relevant literature on the reduction or concentration of AFs through cooking techniques.
  • Figure/Table Captions: Ensure all acronyms (e.g., MOE, HQ, UB, LB) are defined on first use in each table.
  • References: Citations are up to date and appropriate, but reference formatting should be aligned with the journal’s final style.
  • Data Access: Consider uploading the raw data tables (e.g., sample-level AFT concentrations) to a public repository for transparency and reuse.

Comments on the Quality of English Language

Minor English editing

Author Response

Thank you so much for your comments and guidance to help us improve the manuscript.

we have addressed your comments in detail as the attached.

Best regards

Round 2

Reviewer 1 Report

Comments and Suggestions for Authors

The authors replied to the comments.

Reviewer 2 Report

Comments and Suggestions for Authors

The authors have greatly improved the manuscript in response to the referees' suggestions. Hence, I suggest acceptance in its current form.